# Health Technology Assessment of Different Glucosamine Formulations and Preparations Currently Marketed in Thailand

**DOI:** 10.3390/medicines10030023

**Published:** 2023-03-08

**Authors:** Olivier Bruyère, Johann Detilleux, Jean-Yves Reginster

**Affiliations:** 1Division of Public Health, Epidemiology and Health Economics, WHO Collaborating Centre for Epidemiology of Musculoskeletal Health and Ageing, University of Liège, 4000 Liège, Belgium; 2Department of Physical Medicine and Rehabilitation, CHU Liège, 4000 Liège, Belgium; 3Department of Veterinary Management of Animal Resources, University of Liège, 4000 Liège, Belgium; 4Chair for Biomarkers of Chronic Diseases, Department of Biochemistry, College of Science, King Saud University, Riyadh 12211, Saudi Arabia

**Keywords:** osteoarthritis, glucosamine, health technology assessment

## Abstract

Objective: The aim of this study was to evaluate the cost-effectiveness of different glucosamine formulations and preparations used for the management of osteoarthritis in Thailand compared with placebo. Methods: We used a validated model to simulate the individual patient Utility score from aggregated data available from 10 different clinical trials. We then used the Utility score to calculate the quality-adjusted life year (QALY) over 3 and 6 months treatment period. We used the public costs of glucosamine products available in Thailand in 2019 to calculate the incremental cost-effectiveness ratio. We separated the analyses for prescription-grade crystalline glucosamine sulfate (pCGS) and other formulations of glucosamine. A cost-effectiveness cut-off of 3.260 USD/QALY was considered. Results: Irrespective of the glucosamine preparation (tablet or powder/capsule), the data show that pCGS is cost-effective compared with placebo over a 3 and 6 months. However, the other glucosamine formulations (e.g., glucosamine hydrochloride) never reached the breakeven point at any time. Conclusions: Our data show that pCGS is cost-effective for the management of osteoarthritis in the Thai context while other glucosamine formulations are not.

## 1. Introduction

Musculoskeletal conditions are a major public health problem that increases socioeconomic inequalities [1]. One of the major contributor to this is osteoarthritis (OA), and its consequences will be an increasingly important public health concern in the near future [2]. Effective treatments are available to manage OA and, recently, different algorithms or strategies to improve the quality of life of OA patients have been proposed by different international societies such as OsteoArthritis Research International (OARSI) and European Society For Clinical And Economics Aspects Of Osteoporosis Osteoarthritis And Musculoskeletal Diseases (ESCEO) [3,4]. In the last one, the use of symptomatic slow acting drugs is proposed as the background treatment on the condition is that the product used has been shown to be effective in well-designed clinical trials. In fact, there are different formulations or preparations of these products as it is the case with glucosamine or chondroitin [5]. Importantly, all have not been shown to be effective but most of them are still used by many patients and doctors even as dietary supplements. Regarding glucosamine, in the scientific literature, there seem to be major differences between the effect of pharmaceutical grade crystalline glucosamine sulfate (pCGS) and the effect of other formulations of glucosamine [6]. 

Of course it is important to allocate the best treatments to patients but in a world with limited health care resources, the cost of the treatment must be taken into account. Consequently, it is important to compare different strategies in terms of cost and effects. This is the principle of economic evaluation that play in addition a growing role in pricing and reimbursement decisions. In Thailand, there are 3 public healthcare schemes [7]. Firstly, Thailand achieved universal health coverage in 2002 through the implementation of the Universal Coverage Scheme (UCS, also known as the 30-Baht Scheme). The UCS, as the main social health insurance program in the country, currently covers approximately 75% (approximately 47 million people) of the entire population, and it accounts for approximately 17% of the country’s total health expenditure. While the Civil Servant Medical Benefit Scheme (CSMBS), as a fringe benefit, provided health insurance to government sector employees, their dependents (e.g., parents, spouses, and children), and retirees. The Social Security Scheme (SSS) was a compulsory health insurance program for private sector employees, and dependents and retirees were not covered by the scheme.

With regard to glucosamine, no studies have ever been conducted to assess the cost-effectiveness of glucosamine in the specific Thai context. In addition, this study would be particularly necessary because of the use of different formulations of glucosamine in Thailand. We previously developed a model to perform economic evaluation from aggregated data of randomized controlled trials published in the field of osteoarthritis [8]. In the present study, we investigated, using this simulation model, the cost-effectiveness of different glucosamine preparations/formulations used in Thailand. 

## 2. Methods

The principle of this analysis is to compare the cost-effectiveness of different glucosamines (against placebo) over a 3-month and a 6-month treatment horizon. In fact, we will assess different glucosamines in terms of their cost (in $) and their effect (in quality-adjusted life year (QALY)). This economic evaluation of treatments usually requires access to individual patient data to calculate the QALY, which was not possible to obtain. Moreover, the Utility score, needed to assess the QALY, is often not reported as such in the clinical trial reports or in publications related to OA.

Consequently, we used a model previously developed to simulate Utility scores from aggregated data obtained in published trials [8]. However, we only included clinical trials having used the WOMAC score as an outcome measure as it is requested to assess the Utility score (see below). We found 10 trials on glucosamine having used the WOMAC as an outcome measure, 4 using pCGS and 6 using other formulations/preparations of glucosamine. 

We had to transform the WOMAC score into Utility score using a previously–validated formula. It is based on a linear regression model based on the age of the patient, the number of years since the OA diagnosis and the three different WOMAC subscales scores [9]. In fact, the team of Grootendorst developed a prediction model to map the WOMAC along with basic demographic and OA disease severity data into Utility scores. According to the proposed formula, the Utility score = 0.5274776 + 0.0079767 × Pain + 0.0065111 × Stiffness −0.0059571 × Function + 0.0019928 × Pain × Stiffness + 0.0010734 × Pain × Function + 0.0001018 × Stiffness × Function − 0.0030813 × Pain² − 0.0016583 × Stiffness² − 0.000243 × Function² + 0.0113565 × Age in years − 0.0000961 × Age in years² − 0.0172294 × Female − 0.0057865 × Years since onset of OA in the study knee + 0.0001609 × Years since onset of OA in the study knee². This formula was used is this study.

The required data were extracted from published articles, after correction for the scales (to be on the scale for WOMAC indexes as the one used in the equation of Grootendorst) and we replaced missing data in the summary statistic of published studies with data from the study used to develop and validate the procedure for which we had a full access to the data. We then simulated a total of 30,000 patients in each study (15,000 glucosamine and 15,000 placebo) using SIMNORMAL procedure of SAS that performs conditional and unconditional simulations for a set of correlated normal or Gaussian random variables. In fact, using means and variances provided in previous studies, data were simulated by repeated random samples from normal distributions.

The utility estimates were used to calculate the QALY using the area-under-the-curve method that is the weighted average of time spent in the study and utility value. Since we included 10 studies with different durations, if more than one study was available for a particular time, each study was weighted according to the number of subjects included in the trial.

We used the 2019 public costs of glucosamine products available in Thailand without adjustment for inflation. In case of different packaging, we used the most economical in terms of cost per month. We also considered the dose of 1500 mg/day as the standard use of glucosamine, regardless of the severity of the disease, as recommended [3]. We considered the price of the product sell to the general public and the price sell to the government sector. The general public price is a listed price which is set by company/manufacture to sell to hospital either government or private clinics. The government sector price is the “median price”. In fact, Thailand’s Ministry of Public Health determines the maximum procurement price or “median price” for each drug identified under the Median Pricing. If manufacturers cannot offer median priced products at the stipulated median price or lower, they will not be allowed to sell them to government hospitals.

Table 1 represents the costs for one month of the different glucosamines (at the dosage of 1500 mg/day) according to their preparation or sales models. For reasons of confidentiality, we have replaced the trade names with a number. Data on the relationship between trade names and numbers can be provided on request.

Using the cost and the QALYs, we calculated the incremental cost-effectiveness ratio (ICER). We separated the analyses for pCGS and other formulations of glucosamine (e.g., glucosamine hydrochloride, N-acetyl glucosamine). We also separated the analyses according to power or tablet/capsule preparation of the different glucosamines. A cost-effectiveness cut-off of 3.260 $/QALY was considered. In fact, in 2007, the Thai subcommittee responsible for the development of national list of essential medicines set a threshold of 100.000 Thai Baht (i.e., 3.260 $ with the exchange rate of 30.678) per QALY gained [10]. At that time, 100.000 Thai Baht was equivalent to 0.8 of the per-capita gross domestic product.

## 3. Results

Using the model, the simulated QALYs showed an improvement of 0.017 after 3 months of treatment with pCGS and 0.0411 after 6 months. On the other side, the simulated data for the placebo showed a decrease in QALYs after 3 and 6 months, with a change of −0.0088 and −0.0121, respectively. Using public prices, a one-month treatment with pCGS costs 27.78 $ with the powder preparation and 27.22 $ with the tablet/capsule preparation. The government sector prices were 7.83 $ and 19.44 $ for one-month powder and tablet/capsule preparations, respectively. The related cost-effectiveness analyses taking into account the threshold of 3.260 $/QALY are reported in Table 2 and show that pCGS was cost-effective whatever the preparation (powder or tablet/capsule).

For the other formulations of glucosamine, the model showed that the QALYs slightly increased both in the glucosamine and in the placebo group. Indeed, the changes observed after 3 of follow-up were 0.0031 for glucosamine and 0.0021 for placebo. After 6 months, the changes were 0.0048 and 0.0072, respectively. The mean public cost of one-month treatment with other formulations of glucosamine was 14.61 $ with the powder preparation and 10.80 $ with the tablet/capsule preparation. The government sector prices were 7.55 $ and 13.43 $ for one-month powder and tablet/capsule preparations, respectively. The cost-effectiveness analyses, reported in Table 2, show that the other glucosamine formulations were not cost-effective considering the threshold of 3.260 $/QALY.

## 4. Discussion

In this study, using a validated model to simulate individual data from aggregated results of OA clinical trials, we showed that different glucosamine formulations could have different cost-effectiveness impacts. In particular, pCGS was shown to be cost-effective in the management of OA in the Thai context, using a country-specific threshold to assess the cost-effectiveness of the interventions.

In a resource-constrained society, economic evaluations are often used as important tool to support the allocation of efficient health care resources. However, for systematic, transparent, and consistent policy decision making, a clear ceiling country specific threshold is needed. Interventions with an ICER below the accepted ceiling threshold could then be considered as cost-effective. However, there is no scientific standard for setting this threshold [11,12]. Various researches in Thailand have been conducted to determine the accepted ceiling threshold considered to be cost-effective [13,14]. In an community-based survey conducted among 1191 Thai respondents who were face-to-face interviewed to elicit his/her health state preference, it was shown, that from a treatment perspective, the mean willingness to pay for a QALY value estimated by the time trade off method ranged from 59,000 to 285,000 Baht [13]. In contrast, the mean willingness to pay for a QALY value in terms of prevention was lower, ranging from 26,000 to 137,000 Baht. The authors also found that gender, household income, and hypothetical scenarios were also significant factors associated with the willingness to pay/QALYs values [13].

However, it is important to note that taking into account all societal values, cost-effectiveness thresholds are not the only aspect that must be taken into consideration [15]. In Thailand, as in other countries, multi-criteria decision analysis is used as a comprehensive methodological approach to health priority setting [16]. However, it has also been noted that although the use of multi-criteria decision analysis improves the rationality, transparency and fairness of the prioritization process is not always easy to judge in the absence of a clear standard for these aspects. Moreover, because of limited resources, the government cannot make all these interventions available free of charge to the population. Either way, ICER analyses are important and could play a role even in the absence of reimbursement, from the patient’s point of view just to get a better idea of the value for money.

This study highlights the importance of the glucosamine formulation to reach the acceptable cost-effective intervention threshold. Glucosamine can indeed be extracted and stabilized by different chemical modifications and it could potentially impact the biological effect. Different formulations of glucosamine tested in various in vitro systems have been shown to have different mechanisms of action [17,18]. In vivo, various formulations of glucosamine have been tested, including glucosamine sulfate and glucosamine hydrochloride, with different results in terms of clinical effectiveness [6]. For example, the independent meta-analysis of Eriksen et al. including 25 trials and a total of 3458 patients, showed that trials using the pCGS product had a superior effect on pain compared to other formulation of glucosamine [19]. This study is one of those used by ESCEO to support its claim for the use of pCGS and to discourage the use of other formulations [3].

We must recognize certain limitations in this study. First, we do not have individual patient data from the 10 studies included in this analysis. However, we used a validated simulation model developed to overcome this particular problem. Second, we only used placebo as comparator in this study and clearly we agree that in the future other health technology assessments will need to be conducted by comparing different more active OA treatments. Third, our cost-effectiveness analyses do not take into account all the costs of managing OA that could be impacted by the intervention, such as the use of other drugs or over-the-counter products, the number of medical visits or other health care utilization. Fourth, we also cannot rule out the possibility that doctors or patients themselves may change the dosage, which could affect the cost or effectiveness of the treatment. Fifth, the response to therapy, and consequently the ICER, may be influenced by a particular factor, such as the stage of the disease. Indeed, even with placebo, the response to treatment can vary according to the different clinical characteristics of the patients [20]. Sixth, we should also point out that the exact composition of the different types of glucosamine is not always clear, especially when using over-the-counter products. Finally, the utility values used to assess QALYs in this study were only indirectly reported because no clinical trials with glucosamine used a direct assessment of utility value.

In conclusion, in the Thai context, the use of pCGS is cost-effective regardless of whether it is used in powder or tablet/capsule preparations. However, other formulations of glucosamine were not cost-effective in all scenarios. These results confirm the importance of the formulation of glucosamine-based products. As there is currently no treatment that can truly cure OA, most of the currently marketed products that aim to reduce the symptoms of OA could be used for a very long period of time. In the future, other health technology assessments with a long-term or even lifelong perspective will be needed.

## Figures and Tables

**Table 1 medicines-10-00023-t001:** Costs ($) for one month of the different glucosamines (at the dosage of 1500 mg/day) according to their preparation or sales models.

	Powder		Tablet/Capusle	
	Public Price	Government Sector Price	Public Price	Government Sector Price
Pharmaceutical Grade Crystalline Glucosamine Sulfate				
1	27.78	7.83	27.22	1944
Other Formulations of Glucosamine				
2	18.87	7.83		
3	18.87	7.83	13.59	13.59
4	16.99	7.83	9.06	9.06
5			2.04	2.04
6	15.67	7.80		
7	18.84	5.85	8.72	8.72
8	25.53	7.83	26.65	19.44
9	7.00	7.83		
10	12.57	4.38	11.01	7.80
11	14.35	7.83	23.69	19.44
12	6.78	7.32		
13	15.10	7.83		
14			11.07	19.44
15			12.07	19.44
16			9.74	7.80
17	8.67	7.83	7.91	19.44
18	17.50	7.83		
19	8.29	7.83		
20	21.55	7.83		
21			7.92	19.44
22			7.92	19.44
23	13.46	7.83		
24	12.09	7.83	7.37	7.37
25	8.80	7.83	6.80	6.80
26			8.48	19.44
27	15.61	7.83		
28	12.46	7.83		
29			9.69	9.69
30	17.77	7.83		

**Table 2 medicines-10-00023-t002:** Incremental cost-effectiveness ratio (ICER), in $/QALY, of different glucosamine formulations and preparations according to their sales strategies at different time-points.

	Public PricE		Government Sector Price	
	At 3 Months	At 6 Months	At 3 Months	At 6 Months
ICER of crystalline glucosamine sulfate in powder vs. placebo	3230 USD/QALY	3133 USD/QALY	910 USD/QALY	883 USD/QALY
ICER of crystalline glucosamine sulfate in tablet/capsule vs. placebo	3165 USD/QALY	3069 USD/QALY	2260 USD/QALY	2192 USD/QALY
ICER of other glucosamine formulations in powder vs. placebo	42830 USD/QALY	Dominated by placebo	22650 USD/QALY	Dominated by placebo
ICER of other glucosamine formulations in tablet/capsule vs. placebo	32400 USD/QALY	Dominated by placebo	40290 USD/QALY	Dominated by placebo

## Data Availability

The datasets used and/or analyzed during the current study are available from the corresponding author on reasonable request.

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
