# Peer review of "Health Technology Assessment of Different Glucosamine Formulations and Preparations Currently Marketed in Thailand"

_medicines, 2023, doi:10.3390/medicines10030023_

Round 1

Reviewer 1 Report

The paper submitted by Olivier Bruyère and co-workers presents a series of substantive and well-intended different glucosamine formulations and preparations used for the management of osteoarthritis. There are some minor suggestions below that should be addressed by authors:

(1) The authors should correct typing errors throughout the manuscript, for the instance, affiliation 1.

(2) Corresponding authors should be mentioned.

(3) Please provide keywords.

(4) In table1. should be remodify and provide with detailed descriptions.

(5) Please define OARSI and ESCEO.

(6) Author contribution should be added, the authors should follow the template file to remodify the manuscript (https://www.mdpi.com/journal/medicines/instructions).

Author Response

Reviewer 1

The paper submitted by Olivier Bruyère and co-workers presents a series of substantive and well-intended different glucosamine formulations and preparations used for the management of osteoarthritis. There are some minor suggestions below that should be addressed by authors:

(1) The authors should correct typing errors throughout the manuscript, for the instance, affiliation 1.

Authors: We have fully reviewed our text for typing errors.  We thank the reviewer for this important comment.

(2) Corresponding authors should be mentioned.

Authors: we agree and we have now highlighted that Olivier Bruyère is the corresponding author.

(3) Please provide keywords.

Authors: We have now included three keywords: osteoarthritis; glucosamine; health technology assessment.

(4) In table1. should be remodify and provide with detailed descriptions.

Authors: we agree and we have modified the Table. Moreover, we have also added a new table, as requested by reviewer 2.  Our old Table 1 is now Table 2 and new information is also provided in the new Table 1.

(5) Please define OARSI and ESCEO.

Authors: it is an important point and we have defined OARSI and ESCEO.

(6) Author contribution should be added, the authors should follow the template file to remodify the manuscript (https://www.mdpi.com/journal/medicines/instructions).

Authors: we have added the contributions of all authors.

Reviewer 2 Report

I have several major comments that need to be addressed.

1. Abstract

- The comparator and the study perspective should be specified (the comparators were only glucosamine formulations and preparations, or included placebo)

- The time horizon was not consistent between lines 17 and 22

2. Introduction

An introduction relating to the healthcare system in Thailand for those who may not be acquainted with it should be provided

- Explain why the results from the previous study can not be applied to Thailand though the costs from that study from the IMS health data, and you need to conduct this study with the same study design with a bit of difference price of glucosamine. With different prices of glucosamine, does the author intend to submit more papers with different prices for each country?

- The introduction should describe clearly the gaps in the research so you are writing the paper

3. Methods

- Similar to the abstract, the comparators and the study perspective should be specified (the comparators were only glucosamine formulations and preparations, or included placebo)

- Indicate specifically the framework duration of this analysis.

-  What is the difference between the general public and the price sell to the government sector? Is there reimbursement in the estimation of costs? 

- Line 88 “data not shown – available on request”, but the cost is one of the important factors that might affect the result; please provide in detail how the prices of glucosamine were selected and the range of prices. 

- Are there different patients with different levels of OA that led to different prescriptions? If not, explain why.

- What are other glucosamine formulations, please provide them in detail because these supplements aren't considered interchangeable.

- Why only the drug cost of glucosamine was estimated? Are there any difficulties during conducting the research

- The sensitivity analysis is an important assessment, which must be added.

- Provide the reference for choosing the cut-off point (line 95)

- Glucosamines need at least some months to create effectiveness, so how is the QALY when drugs have not worked yet? Are utilities different between stages of OA?

- Please add more information about the SIMNOMAL procedure of SAS.

4. Results

- The total costs and QALYs of each glucosamine and placebo should be displayed in the Table.

- Though the title mentioned the impact of clinical effect and selling price, the results have not shown anything relating to these problems

- ICER should be mentioned specifically which comparators (even in the Table). it is difficult to follow when ICER of which comparison pair was being evaluated (with placebo or with other glucosamines)

- The author indicated that utility scores were affected by the age and the number of years since the OA diagnosis and the three WOMAC subscales scores; whether these factors affected the results of ICER? 

- Sensitive analysis needs to be included in the result section

- Glucosamine is a supplement and needs to be used in the long term; in case of using longer, whether glucosamine is cost-effective? More scenarios need to be analyzed (e.g., long life)

5. Discussion

- Discussion, line 140, provide the reference

- A discussion that explained why only drug cost was estimated is necessary. 

- Why don’t you compare the glucosamine in the liquid, chewable formulations? The powder formulation may not be chosen by the patients by the inconvenience

- Similarly, though the title mentioned the impact of clinical effect and selling price, especially the selling price, the discussion had not shown anything implications or solutions to solve this problem.

- Though the other glucosamine formulations were not indicated specifically, many trials showed that glucosamine sulfate was the best form in efficacy and safety. More discussions about choosing the comparators are necessary.

6. Format

Many typos must be checked. E.g., 

- many double spaces, e.g, line 44 “ Consequently”, line 56 “ Moreover”

- wrong number, e.g., line 69 “+.0065111” 

- "cost effective" changed to "cost-effective"

7. References

- Though the author had many quality studies, the self-citation is too much, the authors should update more other quality studies.

Author Response

Reviewer 2

I have several major comments that need to be addressed.

  1. Abstract

- The comparator and the study perspective should be specified (the comparators were only glucosamine formulations and preparations, or included placebo).

Authors: It is an important point and we have added that the comparator was the placebo.

- The time horizon was not consistent between lines 17 and 22

Authors: We agree and the time horizon has been harmonized.

  1. Introduction

- An introduction relating to the healthcare system in Thailand for those who may not be acquainted with it should be provided

Authors: We fully agree with the reviewer and some information regarding the healthcare system in Thailand has been provided. More particularly, we have added this paragraph: “In Thailand, there are 3 public healthcare schemes. Firstly, Thailand achieved universal health coverage in 2002 through the implementation of the Universal Coverage Scheme (UCS, also known as the 30-Baht Scheme). The UCS, as the main social health insurance program in the country, currently covers approximately 75% (approximately 47 million people) of the entire population, and it accounts for approximately 17% of the country’s total health expenditure. While the CSMBS, as a fringe benefit, provided health insurance to government sector employees, their dependents (e.g., parents, spouses, and children), and retirees. The SSS was a compulsory health insurance program for private sector employees, and dependents and retirees were not covered by the scheme.” We have also added a reference : https://www.ncbi.nlm.nih.gov/pmc/articles/PMC5104696/

- Explain why the results from the previous study can not be applied to Thailand though the costs from that study from the IMS health data, and you need to conduct this study with the same study design with a bit of difference price of glucosamine. With different prices of glucosamine, does the author intend to submit more papers with different prices for each country?

Authors: This is an important point raised by the reviewer. In fact, this study is particularly necessary because of the use of different formulations of glucosamine, which were not evaluated in our initial model. This information was added in our Introduction.

- The introduction should describe clearly the gaps in the research so you are writing the paper

Authors: We agree with the reviewer and gaps in the research have been added. We have added this text: “With regard to glucosamine, no studies have ever been conducted to assess the cost-effectiveness of glucosamine in the specific Thai context. In addition, this study is particularly necessary because of the use of different formulations of glucosamine, which were not evaluated in our initial model.”

  1. Methods

- Similar to the abstract, the comparators and the study perspective should be specified (the comparators were only glucosamine formulations and preparations, or included placebo)

Authors: We agree with the reviewer and comparator has been added.

- Indicate specifically the framework duration of this analysis.

Authors: The framework duration has been added.

-  What is the difference between the general public and the price sell to the government sector? Is there reimbursement in the estimation of costs?

Authors: We fully agree that our initial sentence was not clear at all.  The general public price is listed price which is set by company/manufacture to sell to hospital either government or private and clinic.       Government sector price is median price (MPP). Thailand’s Ministry of Public Health determines the maximum procurement price or “median price” for each drug identified under the Median Pricing. If manufacturers cannot offer median priced products at the stipulated median price or lower, they will not be allowed to sell them to government hospitals. We have now included the information in our Methods section.

- Line 88 “data not shown – available on request”, but the cost is one of the important factors that might affect the result; please provide in detail how the prices of glucosamine were selected and the range of prices.

Authors: We agree with the reviewer that it is a very important point.  We have added a new Table (i.e. Table 1) with the costs for one month of the different glucosamines (at the dosage of 1500 mg/day) according to their preparation or sales models.  For reasons of confidentiality, we have replaced the trade names with a number.  Data on the relationship between trade names and numbers can be provided on request. 

- Are there different patients with different levels of OA that led to different prescriptions? If not, explain why.

Authors: This an important point raised by the reviewer.  According to experts, the dose of 1500mg per day is recommended regardless of the severity of the disease.  This has been included in the Methods section. However, we cannot exclude the possibility that doctors or patients themselves may change the dosage.  This has been added to the discussion section.

- What are other glucosamine formulations, please provide them in detail because these supplements aren't considered interchangeable.

Authors: This is a complex point, as the reviewer points out.  The problem is that the exact composition of some of the over-the-counter glucosamine products is not clear.  However, we have provided some information in the Methods section and we have also discussed this point in the Discussion section as a potential limitation of our study.

- Why only the drug cost of glucosamine was estimated? Are there any difficulties during conducting the research?

Authors: We are not sure we understand the reviewer's point.  We assume that the reviewer is asking why all other costs of osteoarthritis were not taken into account.  In fact, it was not possible in this particular study and we have already included this sentence in the discussion section: ”our cost-effectiveness analyses do not take into account all the costs of managing OA that could be impacted by the intervention, such as the use of other drugs or over-the-counter products, the number of medical visits or other health care utilization.”

- The sensitivity analysis is an important assessment, which must be added.

Authors: In the particular context of this study, no sensitivity analysis was carried out because we have already done different analyses according to the formulations of glucosamines and sales prices.  Therefore, we do not have a sensitivity analysis to propose.

- Provide the reference for choosing the cut-off point (line 95)

Authors: We agree and have provided a reference for this particular statement.

- Glucosamines need at least some months to create effectiveness, so how is the QALY when drugs have not worked yet? Are utilities different between stages of OA?

Authors: Again, an important point raised by the reviewer.  The QALY calculated in this trial is an average of all patients, regardless of their response to glucosamine or placebo.  It is likely that the response to therapy may be influenced by a particular factor, such as the stage of the disease. Incidentally, even with placebo, the response can vary according to the different clinical characteristics of the patients.  We have included this point in the Discussion section and added this reference: Wen X, Luo J, Mai Y, et al. Placebo Response to Oral Administration in Osteoarthritis Clinical Trials and Its Associated Factors: A Model-Based Meta-analysis. JAMA Netw Open. 2022;5(10):e2235060. doi:10.1001/jamanetworkopen.2022.35060

- Please add more information about the SIMNOMAL procedure of SAS.

Authors: We agree that our text was not clear and we have added the information that using means and variances provided in previous studies, data were simulated by repeated random samples from normal distributions.

  1. Results

- The total costs and QALYs of each glucosamine and placebo should be displayed in the Table.

Authors: As requested, we have included a new table showing the cost of each glucosamine.  The QALYs of the different formulations are also given in the text.  However, it is not possible to calculate a QALY for each product because, by definition, our methodology uses aggregated data from randomized controlled trials.

- Though the title mentioned the impact of clinical effect and selling price, the results have not shown anything relating to these problems

Authors: We agree that the title of our article is not very accurate and we have changed it.

- ICER should be mentioned specifically which comparators (even in the Table). it is difficult to follow when ICER of which comparison pair was being evaluated (with placebo or with other glucosamines)

Authors: We fully agree with the reviewer that our Table was not clear.  We have modified it.

- The author indicated that utility scores were affected by the age and the number of years since the OA diagnosis and the three WOMAC subscales scores; whether these factors affected the results of ICER?

Authors: As a matter of fact, our text was not clear enough.  This is indeed the transformation of the WOMAC into Utility score that is influenced by the age of the patient, the number of years since the OA diagnosis and the three different WOMAC subscales scores.  We have now made our text clearer. However, it a possible that the ICER may be influenced by a different clinical characteristics of the patients.  It has been added in the Discussion section.

- Sensitive analysis needs to be included in the result section

Authors: As previously discussed, no sensitivity analysis was carried out because we have already done different analyses according to the formulations of glucosamines and sales prices.  Therefore, we do not have a sensitivity analysis to propose.

- Glucosamine is a supplement and needs to be used in the long term; in case of using longer, whether glucosamine is cost-effective? More scenarios need to be analyzed (e.g., long life)

Authors: This is an important point, but unfortunately we do not have data on lifetime use.  However, it is of paramount importance and we have added this point in our discussion section. We have added this sentence: "As there is currently no treatment that can truly cure OA, most of the currently marketed products that aim to reduce the symptoms of OA could be used for a very long period of time.  In the future, other health technology assessments with a long-term or even lifelong perspective will be needed.”

  1. Discussion

- Discussion, line 140, provide the reference

Authors: References has been added. Thank you.

- A discussion that explained why only drug cost was estimated is necessary.

Authors: In fact, we acknowledge that this is a major limitation of our study, but it is not possible to take it into account in this particular analysis due to its particular methodology.  However, we have included this point in the limitations in the Discussion section.

- Why don’t you compare the glucosamine in the liquid, chewable formulations? The powder formulation may not be chosen by the patients by the inconvenience

Authors: To our knowledge, there are no liquid or chewable formulations of glucosamine.

- Similarly, though the title mentioned the impact of clinical effect and selling price, especially the selling price, the discussion had not shown anything implications or solutions to solve this problem.

Authors: As you previously suggested, we have changed the title, which did not reflect our study objectives very well.

- Though the other glucosamine formulations were not indicated specifically, many trials showed that glucosamine sulfate was the best form in efficacy and safety. More discussions about choosing the comparators are necessary.

Authors: We agree that in the future other health technology assessments will need to be conducted by comparing different active treatments.  This was not possible due to the complexity of the current methodology, but we have included this important point in the discussion section.

  1. Format

Many typos must be checked. E.g.,

- many double spaces, e.g, line 44 “ Consequently”, line 56 “ Moreover”

- wrong number, e.g., line 69 “+.0065111”

- "cost effective" changed to "cost-effective"

Authors: We have fully reviewed our text for typing errors.  We thank the reviewer for this important comment.

  1. References

- Though the author had many quality studies, the self-citation is too much, the authors should update more other quality studies.

Authors: We agree with the reviewer and some references of our group have been removed while references from other groups have been added.

Round 2

Reviewer 2 Report

- Please define SSS and CSMBS

-  The author should add more information about the year of price unit taken. Are there any adjustments for inflation?

- Table 1. add currency unit

- Table 1. From 1 to 30, please specify which was pCGS and which were other formulations of glucosamine (e.g., 121 glucosamine hydrochloride, N-acetyl glucosamine) as mentioned

Author Response

- Please define SSS and CSMBS

Authors :

SSS stands for Social Security Scheme and CSMBS for Civil Servant Medical Benefit Scheme.  It has been added in the manuscript. 

-  The author should add more information about the year of price unit taken. Are there any adjustments for inflation?

Authors: We used the 2019 public costs of glucosamine products available in Thailand but no adjustement for inflation where undertaken.  It has been added in the manuscript.

- Table 1. add currency unit

Authors: the cost is in $.  It has been added in the title of the Table 1. 

Table 1. From 1 to 30, please specify which was pCGS and which were other formulations of glucosamine (e.g., 121 glucosamine hydrochloride, N-acetyl glucosamine) as mentioned

Authors: we have added the information.  In fact, the 29th firsts are “other glucosamine formulation” and the 30th is crystalline glucosamine sulfate.  For clarity, we have put crystalline glucosamine sulfate as the number 1 and the other formulation as number 2 to 30.